# Organ transformation by environmental disruption of protein integrity and epigenetic memory in *Drosophila*

Orli Snir[1‡]*, Michael Elgart[1‡], Yulia Gnainsky[1], Moshe Goldsmith[1], Filippo Ciabrelli[2], Shlomi Dagan[1], Iris Aviezer[1], Elizabeth Stoops[1], Giacomo Cavalli[2], Yoav Soen[1]*

**1** Department of Biomolecular Sciences, Weizmann Institute of Science, Rehovot, Israel, **2** Institute of Human Genetics, UMR9002 CNRS, University of Montpellier, Montpellier, France

‡ These authors share first authorship on this work.
* osnir@rockefeller.edu (OS); yoavs@weizmann.ac.il (YS)

**Data Availability Statement:** All ChIP-seq and RNA-seq datasets are available from the SRA database BioProject PRJNA466151 Accession: PRJNA466151 ID: 466151. The mass

## Abstract

Despite significant progress in understanding epigenetic reprogramming of cells, the mechanistic basis of "organ reprogramming" by (epi-)gene–environment interactions remained largely obscure. Here, we use the ether-induced haltere-to-wing transformations in *Drosophila* as a model for epigenetic "reprogramming" at the whole organism level. Our findings support a mechanistic chain of events explaining why and how brief embryonic exposure to ether leads to haltere-to-wing transformations manifested at the larval stage and on. We show that ether interferes with protein integrity in the egg, leading to altered deployment of Hsp90 and widespread repression of Trithorax-mediated establishment of active H3K4me3 chromatin marks throughout the genome. Despite this global reduction, Ubx targets and wing development genes preferentially retain higher levels of H3K4me3 that predispose these genes for later up-regulation in the larval haltere disc, hence the wing-like outcome. Consistent with compromised protein integrity during the exposure, the penetrance of bithorax transformations increases by genetic or chemical reduction of Hsp90 function. Moreover, joint reduction in *Hsp90* and *trx* gene dosage can cause bithorax transformations without exposure to ether, supporting an underlying epistasis between *Hsp90* and *trx* loss-of-functions. These findings implicate environmental disruption of protein integrity at the onset of histone methylation with altered epigenetic regulation of developmental patterning genes. The emerging picture provides a unique example wherein the alleviation of the Hsp90 "capacitor function" by the environment drives a morphogenetic shift towards an ancestral-like body plan. The morphogenetic impact of chaperone response during a major setup of epigenetic patterns may be a general scheme for organ transformation by environmental cues.

## Introduction

Determination of cell and tissue identities in flies is established during embryonic development and maintained by epigenetic means, particularly by the Polycomb and Trithorax

spectrometry proteomics data have been deposited to the ProteomeXchange Consortium via the PRIDE partner repository with the dataset identifier PXD050958.

**Funding:** This work was supported by the Sir John Templeton Foundation (40663 to YS; 764 61122 to YS), a research grant from the Fannie Sherr Fund (to YS), and a research grant from the Estate of Benjamin Carasso (to YS). The funders didn't play any role in the study design, data collection and analysis, decision to publish, or preparation of the manuscript.

**Competing interests:** The authors have declared that no competing interests exist.

**Abbreviations:** CD, circular dichroism; ChIP, chromatin immunoprecipitation; DDA, data-dependent acquisition; DTT, dithiothreitol.

systems [1,2]. Early embryonic exposure to biotic and abiotic environmental stimuli (e.g., chemicals, heat, unusual diets, crowding, predation, electromagnetic fields, and many more) can interfere with tissue identities and induce homeotic transformations in a wide range of species [3–7]. Hallmark examples include environmental induction of haltere-to-wing phenocopies [8–10], antenna-to-leg, leg-to-wing, and eye-to-wing transformations by disruptions of primordial organs and boundaries [11,12], as well as limb malformations and/or organ failures by embryonic exposure to various substances (e.g., thalidomide) at a sensitive time window of development in human and other mammals [13–15].

Some of these induced abnormalities resemble spontaneous disfigurements caused by mutations in key developmental regulators, particularly homeotic genes (e.g., Hox genes), epigenetic regulators (e.g., Polycomb and Trithorax-group genes), and secreted morphogens (e.g., signals from the bone morphogenetic proteins and secreted proteins from the Wnt and Sonic hedgehog pathways) [16–18]. The frequency and severity of environmentally-induced abnormalities can also be enhanced by loss-of-function mutations in various patterning genes, suggesting that the respective gene function contributes to suppression of the induced malformation [16,19].

In a landmark experiment, Waddington showed that environmental induction of gross morphological changes can be rapidly enhanced and "stabilized" (genetically assimilated) by successive exposures and selections of phenotypic individuals over a few generations [10]. Stabilization of altered phenotypes was indicated by spontaneous transformations (generation of transformed individuals without exposure to the environmental inducer). The validity and generality of this genetic assimilation were supported by: (i) identified accumulation of alleles contributing to stronger induction and/or generation of spontaneous transformations; (ii) demonstration of similar effects in other scenarios of environmentally-induced transformations; and (iii) evidence for potential stabilization by epigenetic means [19–22].

Combining genetic (and/or epigenetic) assimilation of environmental induction with resemblance to morphogenetic phenotypes of mutations in patterning genes gave rise to a nontraditional view of the environment as a driver of rapid morphological diversification (in addition to its contribution to selection) [3,23–30]. The potentially profound impact on evolution, however, often comes at the expense of generating deformed individuals ("hopeful monsters"). Suppressing these harmful changes and promoting reversal toward normal development requires detailed understanding of the teratologic process [3,31]. While effector genes have been identified in various cases, mechanistic understanding of the chain of events connecting the respective environmental disruptor with stage-specific changes that lead to disfigurement is generally lacking. Here, we use the classic example of bithorax induction as a model for investigating how brief exposure to ether during early embryogenesis modifies the adult body plan. The resemblance to phenotypes of *Ubx* and *trx* mutations [16,18] suggests that environmental induction of bithorax phenocopies may be mediated by reduced function of *Ubx* and/or *trx*. However, it is not clear how the function of these (and/or other) genes mediates the stage-dependent response to ether, why this induction requires exposure in a specific time window, and why a similar transformation is observed in response to heat exposure at the same time window [32].

By analyzing stage-dependent effects of exposure to ether, we provide evidence for a mechanistic chain of events connecting brief embryonic exposure to ether with organ transformation manifested at the larval stage and on. We show that ether disrupts the eggshell and interferes with native protein folding in the embryo. The induction of proteotoxic stress alters the profile of deployments of Hsp90 toward its clients, including Trx. This, in turn, leads to a decrease in Trx function and repression of H3K4 tri-methylation at a critical stage of development. We further found that the repression of active chromatin marks (H3K4 tri-methylation) is less

pronounced in actively transcribed genes, including Ubx targets and wing development genes. The differentially higher postexposure levels of active chromatin marks in wing genes predisposes these genes for later up-regulation in the larval haltere disc. The joint contribution of the interaction between proteotoxic stress and epigenetic patterning to the induction of bithorax phenocopies was supported by impacts of reduced function of Hsp90. We found that the induction of bithorax phenocopies is enhanced by chemical or genetic reduction of Hsp90 and that joint reduction of *Hsp90* and *trx* gene dosage causes spontaneous bithorax transformations, implicating the interaction between Hsp90 and Trx loss-of-functions in causing bithorax transformations. This is consistent with a reported dependence of Trx on Hsp90 function [33]. Altogether, these findings link the proteotoxic stress of ether to excess demand for Hsp90 function and disruption of the initial epigenetic patterning which predisposes wing genes for later up-regulation in the haltere disc. Notably, this chain of events can also account for the reported induction of bithorax phenocopies in response to brief exposure to heat at this critical stage in development [32,34].

## Results

### Exposure to ether disrupts protein structure and alters the deployment of Hsp90 clients in the early embryo

To investigate how embryonic exposure leads to homeotic transformations that are manifested at a later stage, we established effective conditions for bithorax induction by 30-min exposure of early-stage embryos to ether vapor (Fig 1A). To quantify the reaction, we scored the number and fraction of adult flies (and failed-to-eclose pupae) exhibiting bithorax phenocopies. The latter were defined as any morphological abnormality in the third thoracic segment that increases resemblance to the second thoracic segment [10,35–37] (Fig 1B). Bithorax phenocopies were induced in approximately 9% of the individuals without significant reduction in the number of pupae (Fig 1C and 1D). We then sought to investigate how exposure to ether causes these transformations. Since ether is an organic solvent, we suspected that its vapor may have a denaturing impact which could disrupt protein function and induce proteotoxic stress. To investigate this possibility, we first analyzed the effect of ether on the integrity of the eggshell. Inspection of embryos that were exposed for 2 h to the vapor revealed a gross increase in egg clarity (Fig 1E), suggesting elevated egg permeability and lipid precipitation. This was confirmed by increased penetration of the nucleic-acid stain, Acridine Orange [38], into eggs that were exposed to ether vapor (Fig 1F). To determine if the exposure to ether vapor affects the integrity of proteins in the embryo, we analyzed lysates of embryos by (Far-UV) circular dichroism (CD) [39,40]. Comparison of extracts from (in vivo) exposed versus non-exposed embryos revealed a significant decrease in the degree of light polarization produced by proteins in the sample (Fig 1G), suggesting a widespread impact on protein secondary structures, specifically α-helices. Staining with the protein conformation-sensitive probe, 8-anilino-1 naphthalene sulfonate (ANS) [41], revealed concordant differences between extracts from exposed versus non-exposed embryos (Fig 1H). Notably, the direction of change in response to ether was the same as in heat denaturation. Independent analysis of the effect of ether on fluorescence intensity in live, Histone-RFP tagged (His2Av-mRFP1) embryos revealed a significant reduction of RFP intensity in ether-exposed versus control embryos (Fig 1I). This reduction occurred without a change in His2Av mRNA (S1 Spreadsheet), indicating post-transcriptional impact of ether on the His2Av-RFP protein. These findings show that exposure to ether vapor compromises the integrity of the eggshell and causes disruptions in protein structure and function.

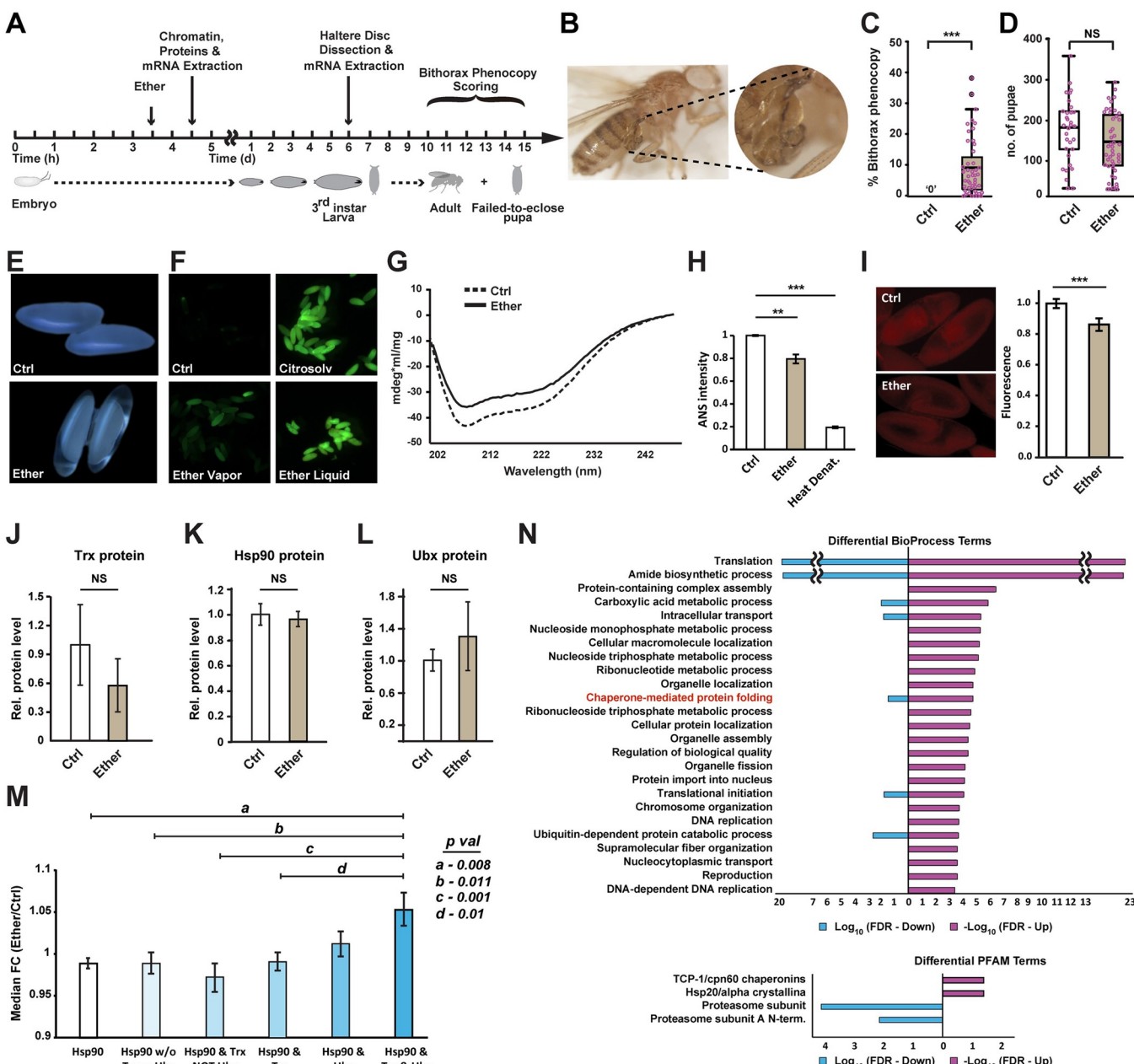

**Fig 1. Ether induces proteotoxic stress, leading to a decrease in Trx protein and an increase in Hsp90 clients that are joint targets of Trx and Ubx. (A)** Flowchart of experimental procedures and measurements. **(B)** Representative image of a severe transformation in an adult fly (*yw* line) that was exposed to ether during early embryogenesis. **(C)** Percentage of individuals exhibiting bithorax phenocopies (including abnormal pupae that failed to eclose). $p < 0.001$, Mann–Whitney. **(D)** Numbers of pupae formed with and without exposure to ether. $p > 0.05$, Student's $t$ test. **(E)** Effect of ether vapor on egg perimeter and transparency. Representative images (10×) of untreated eggs (top) and eggs that were exposed to ether vapor for 2 h (bottom). **(F)** Representative images of Acridine Orange-stained *yw* embryos for the following cases: no treatment (up left), 5-min immersion in Citrasolv solution [46] (up right), 5-min immersion in ether liquid (bottom right), and 1.5-h exposure to ether vapor (bottom left). **(G)** Circular dichroism (Far-UV) spectra of proteins extracted from *yw* embryos that were exposed or not exposed to ether vapor (solid and dotted lines, respectively). Displayed spectra (in [millidegrees*milliliter/milligram]) represent the average of 3 independent measurements. $p < 0.05$, Student's $t$ test. **(H)** Fluorescence intensity of lysates of embryos (*yw* line) stained with 8-anilino-1-naphthalene sulfonate (ANS), a fluorescent probe for protein conformational changes. Shown for untreated embryos (Ctrl), ether-exposed embryos (Ether), and embryos that were denatured at 80°C. Mean intensity ± SE, based on 3 biological replicates. **$p < 0.01$, ***$p < 0.001$, one-way ANOVA following Dunnett's test. **(I)** Representative images (left) and average fluorescence intensity (right) in His2Av-mRFP1-tagged embryos, with and without exposure to ether vapor. Mean intensity per embryo ± SE, $n = 32$ (Ctrl), $n = 29$ (Ether). ***$p < 1E-4$, Mann–Whitney test. **(J)** Trx protein levels following exposure vs. no exposure control, as determined by mass spectrometry analysis. Mean levels (relative to the average in untreated control) ± SE, based on 3 replicates of ether-exposed and control embryos. $p > 0.05$, Mann–Whitney. **(K)** Same as (J) for the *Drosophila* Hsp90 (Hsp83) protein level. Mean ± SE, $n = 3$. $p > 0.05$, Student's $t$ test. **(L)** Western blot analysis of Ubx protein levels in exposed embryos and non-exposed control. Mean ± SE, $n = 3$. $p > 0.05$, Student's $t$ test. **(M)** Median

fold-change (ether/ctrl) of protein levels in the following subsets of Hsp90 clients (left to right): All clients ($n = 280$), Hsp90 w/o Trx or Ubx targets ($n = 197$), Hsp90 and Trx but not Ubx targets ($n = 52$), Hsp90 and Trx targets ($n = 69$), Hsp90 and Ubx targets ($n = 31$), Hsp90 and Trx and Ubx targets ($n = 16$). Based on mass spectrometry analysis with 3 replicates of ether vs. control. Subset-specific median fold-change (±SE) averaged over the proteins in the subset. Note the preferential increase of Hsp90 clients that are targeted jointly by Ubx and Trx. *a*, *b*, *c*, *d*: *p*-values based on Student's *t* test. **(N)** GO enrichment analysis of proteins that increased (red) and decreased by ether, as determined by proteomics profiles of exposed and non-exposed embryos. Based on 3 biological replicates. ** $p < 0.01$, *** $p < 0.001$. The data underlying this figure can be found in S1 Data.

The proteotoxic stress caused by exposure to ether is expected to increase the workload on protein chaperones such as Hsp90, potentially altering their target deployment [42,43]. Moreover, mutations in Trx (a reported client of Hsp90 [33]) generate bithorax transformations [16], suggesting that the ether-induced phenocopies are mediated by impact of the proteotoxic stress on Hsp90 and Trx. To investigate the potential involvement of these genes in the response to ether, we analyzed the effect of ether on the protein levels of Hsp90 and its targets, including Trx. Proteomics analysis of embryos shortly after exposure revealed reduced levels of Trx (Fig 1J) without a change in the protein levels of Hsp90 (Figs 1K and S1). Western blot analysis of Ubx (a downstream target of Trx that specifies haltere fate by repressing the transcription of wing-related genes [18,44,45]) showed that Ubx levels are also unaffected by ether (Figs 1L and S2). In contrast to the lack of change in the levels of Hsp90 and Ubx, the protein levels of Hsp90 clients that are joint targets of Ubx and Trx were significantly elevated compared to other subsets of Hsp90 clients (Fig 1M). The response to ether was therefore accompanied by a differential increase of Hsp90 clients that are likely to favor bithorax transformation. Gene ontology analysis of ether-induced changes in the proteome further showed that elevated proteins are enriched with chaperone genes and decreased proteins are enriched with proteasome genes (Fig 1N).

## Bithorax induction is aggravated by reduced function of *trx* and *Hsp90*

To investigate the involvements of Trx and Hsp90 in the induction of bithorax phenocopies, we analyzed the impacts of ether on *trx* and *Hsp90* mutant stocks versus wild type. The penetrance of bithorax phenocopies in ether-exposed, temperature-sensitive $trx^{1-/+}$ and $trx^{1-/-}$ stocks increased with the mutation dosage. Notably, over 20% of $trx^{1-/-}$ individuals exhibited spontaneous transformation without exposure to ether (Fig 2A). A significant increase in penetrance was also noted in *Hsp90* heterozygotes ($Hsp83^{e6A-/+}$) versus wild-type control (Fig 2B). Aggravation was also observed in ether-exposed wild-type embryos that were pretreated immediately after egg deposition with the Hsp90 inhibitor, Geldanamycin (GdA) (Fig 2C). Analysis of $Hsp83^{e6A-/+}/trx^{1-/+}$ double heterozygotes showed that a combined reduction of *trx* and *Hsp90* gene dosage can cause spontaneous transformations even in non-exposed embryos (Fig 2D). This was in notable contrast to lack of spontaneous transformations in *trx* and *Hsp90* single mutants. Taken together with the dependence of *trx* on *Hsp90* function [33], these findings suggest that loss-of-function of *Hsp90* aggravates the bithorax induction by contributing to the reduction of *trx* function.

## Ether suppresses H3K4 trimethylation in the early embryo

To seek additional evidence for joint involvement of *Hsp90* and *trx* loss-of-function in the induction of bithorax phenocopies, we tested if the exposure to ether compromises the activity of the Trx protein. Since Trx is responsible for the tri-methylation of histone H3K4 (H3K4me3) in the early embryo [47–49], we examined changes in histone methylation and gene expression shortly after exposure (Fig 1A). Active and repressive chromatin marks were analyzed by ChIP-seq using antibodies specific for H3K4me3 and tri-methylation of histone

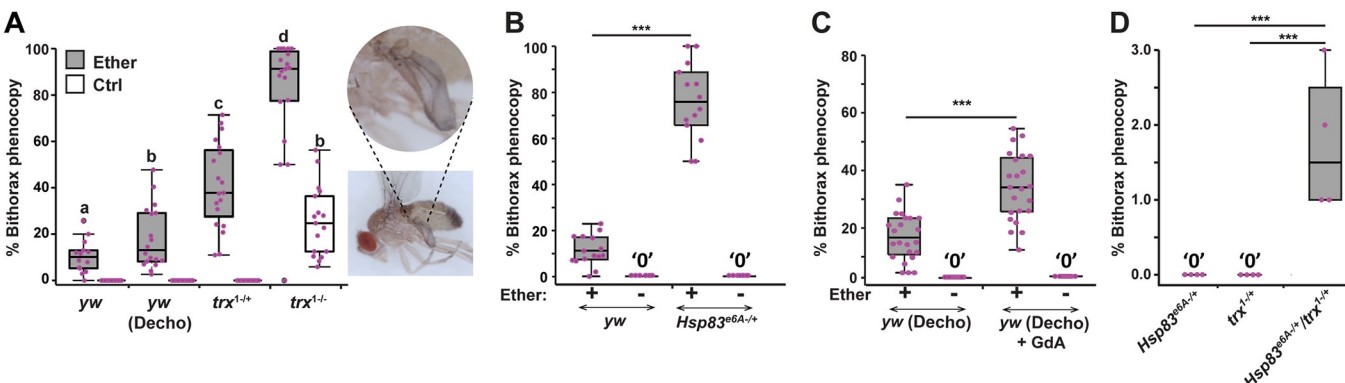

**Fig 2. Epistasis between loss-of-functions of *Hsp90* and *trx*.** (A) Penetrance of bithorax phenocopies in ether-exposed and non-exposed in *trx*[1-/+], *trx*[1-/-] and a wild-type stock (*yw*), with and without egg dechorionation ("Decho") prior to exposure. Two-way ANOVA followed by Tukey HSD test. Groups denoted by different letters (*a*, *b*, *c*, *d*) are significantly different from each other (*p*-values provided in S1 Table). Based on 3 biological replicates pooled. Inset: Exemplary image of ether-induced phenocopies in the *trx*[1] stock. (B) Percentages of flies exhibiting bithorax phenocopies in ether-exposed and non-exposed *Hsp83*[e6A+/-] and wild-type (*yw*) stocks. ***$p$ = 8.4e-11, Student's *t* test. (C) Same as (B) on *yw* background, with and without treatment of dechorionated eggs with the Hsp90 inhibitor, Geldanamycin (*yw* (Decho) + GdA), prior to ether exposure. Based on 3 biological replicates pooled. ***$p$ = 3.7e-07, Student's *t* test. (D) Percentages of *Hsp83*[e6A-/+]/*trx*[1-/+], *trx*[1-/-], and *Hsp83*[e6A+/-] flies exhibiting spontaneous bithorax phenocopies. Based on 4 biological replicates with approximately 100 flies each. ***$p$ < 0.001. The data underlying this figure can be found in S2 Data.

H3K27 (H3K27me3), respectively [50] (Fig 3A and 3B). Comparison of methylation profiles was preceded by global percentile normalization of read counts applied to 100 bp genomic segments of each sample [51]. Differences between normalized counts of exposed ("Ether") versus non-exposed embryos ("Ctrl") revealed a genome-wide (albeit not uniform) decrease of H3K4me3 levels, without a substantial change in H3K27me3 (Fig 3A versus Fig 3B; S3 and S4A Figs). Predominant suppression of H3K4me3 was also reflected by site-specific ratio of H3K4me3 to H3K27me3, determined by integrating normalized counts over each gene region and dividing the region-specific H3K4me3 level by the respective level of H3K27me3. The H3K4me3/H3K27me3 ratio in ether-exposed versus control embryos was lower in 88% of the genes and higher in only 5% of the genes (Fig 3C). Unlike the extensive suppression of H3K4me3, the immediate transcriptional response was very mild (Figs 3D and S4B–S4E and S1 Spreadsheet). Moreover, the up-regulated genes were not enriched with wing genes (Fig 3E), suggesting that the induced predisposition towards wing might be primarily specified by site-specific changes in histone methylation. This was supported by significantly higher abundance of H3K4me3 at loci of wing genes (Figs 3F, S4F, and S4G), as well as by enrichment of wing genes in genomic regions exhibiting the highest retention of H3K4me3 following exposure to ether (Fig 3G and S2 Table). The levels of retention of H3K4me3 were significantly correlated with the levels of mRNA of the respective gene (Figs 3H and S5A–S5D), suggesting that active expression of wing genes during the exposure contributes to higher levels of H3K4me3 in wing genes versus bulk. Given the reported contribution of H3K4me3 to long-term stability of active state of transcription [52,53], higher levels of H3K4me3 in wing genes at the end of exposure could predispose these genes for differentially higher expression at later stages.

## Ether up-regulates wing-related Trx-Ubx targets in the haltere disc

The higher penetrance of bithorax phenocopies in *trx* mutants versus wild type (Fig 2A) suggested a causal link between the repressive effect of ether on the (*trx-mediated*) H3K4me3 in the embryo and the haltere-to-wing transformation in the larva. To investigate this possibility, we analyzed mRNA changes in the haltere discs of third instar larvae from ether-exposed versus non-exposed wild-type and *trx* mutant stocks (S6A Fig and S2 Spreadsheet). RNA-seq

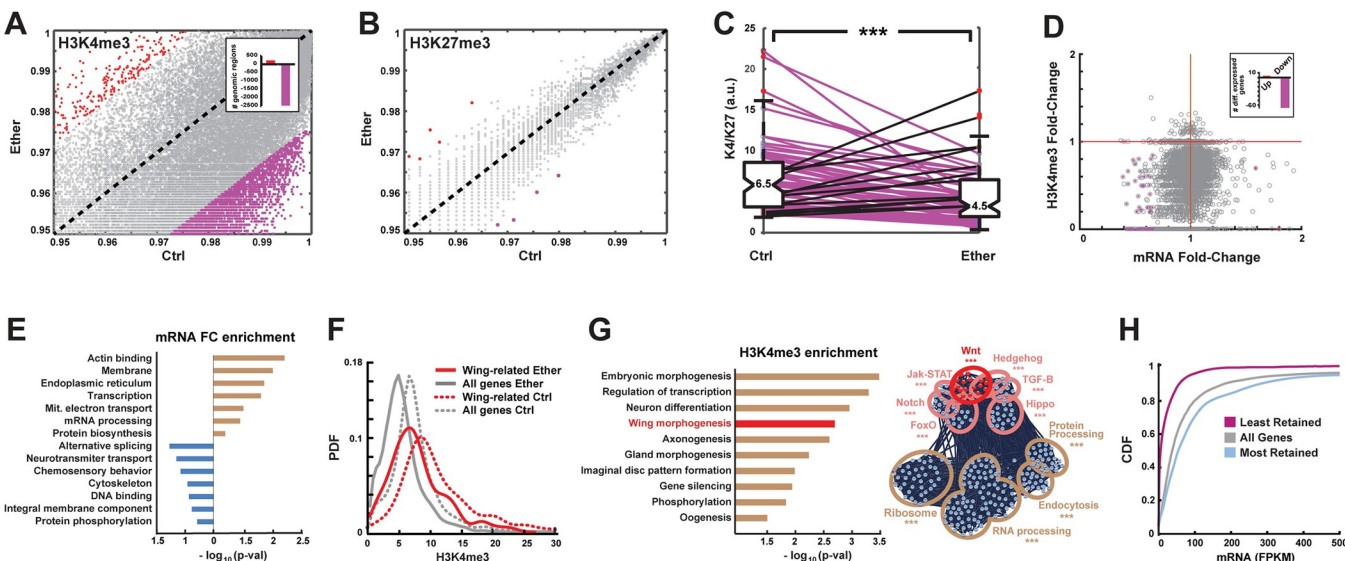

**Fig 3. Ether suppresses H3K4me3, while retaining relatively higher levels at wing gene loci. (A)** Percentile-normalized numbers of H3K4me3 reads per 100 bp in ether-exposed ("Ether") vs. non-exposed *yw* embryos ("Ctrl"). Red and purple dots correspond, respectively, to changes above and below 2 standard deviations from the mean. Inset: no. of regions with methylation levels above and below 2 standard deviations from the mean. **(B)** Same as (A) for H3K27me3 reads per 1,000 bp. **(C)** Gene-specific H3K4me3/H3K27me3 ratios in ether-exposed vs. non-exposed embryos. *** $p <$ 1E-19, Wilcoxon signed rank test. **(D)** mRNA fold-change (Ether/Ctrl) vs. H3K4me3 fold-change (Ether/Ctrl). Inset: numbers of differentially expressed genes (fold-change >1.5 and $p <$ 0.05), based on 3 biological replicates. **(E)** Enrichments of GO terms in subsets of genes that are up- (brown) and down-regulated (blue) 1 h after exposure vs. no exposure. Based on the "DAVID" online tool. **(F)** Probability density functions (PDF) of H3K4me3 levels, demonstrating a shift in the distribution of "wing development" genes (red) toward higher levels compared to bulk (gray) in both ether-exposed and control embryos (solid and dotted lines, respectively). *** $p <$ 1E-6. **(G) Left:** Enrichments of GO terms in genomic loci exhibiting the highest retention of H3K4me3 (top 10%) following exposure to ether. Based on the "DAVID" online tool. Gene-specific retention is defined by the ratio between H3K4me3 levels in ether-exposed and non-exposed embryos. **Right:** STRING Network analysis of genes with the highest retention of H3K4me3. Significance levels are indicated in S2 Table. **(H)** Cumulative distribution function (CDF) of mRNA levels, shown for all genes with a detectable level of H3K4me3 (gray) as well as for genes with 10% highest and lowest retention of H3K4me3 (blue and purple, respectively). *** $p <$ 1E-6 corresponds to the difference between the bulk distribution and the distributions for genes with either lowest or highest retention. ChIP and mRNA analyses are based on 2 and 3 biological replicates, respectively.

analysis of dissected discs revealed dosage-dependent induction of wing genes (Fig 4A), including wingless (*wg*), the master regulator of wing development (Fig 4B). Ubx targets and joint targets of Ubx and Trx were also up-regulated, whereas targets of Trx alone were down-regulated (Fig 4C and 4D). The relevance of these changes to the bithorax induction was further highlighted by preferential up-regulation of wing genes that are jointly targeted by Ubx, but not wing genes that are targeted by Trx alone (Fig 4E and 4F). Since *Ubx* is itself a target of Trx and a transcriptional repressor of wing genes in the haltere [44,45], the up-regulation of joint targets of Ubx and Trx is consistent with the induction of wing phenotypes. Notably, despite the significant changes in the levels of their targets, the mRNA levels of *trx* and *Ubx*, as well as the protein levels of Ubx, were not affected by exposure to ether (S6B, S6C, and S7 Figs and S2 Spreadsheet).

## Induction of wing genes in the haltere is reflected by H3K4me3 levels shortly after the exposure

While up-regulation of joint targets of Ubx and Trx in the haltere discs may account for the ectopic wing phenotypes, it is not clear how this up-regulation is promoted by brief exposure to ether during early embryogenesis. To investigate if the induction of wing fates could be specified by the impact of ether on H3K4 trimethylation in the early embryo, we examined correlations between H3K4me3 levels shortly after exposure and mRNA levels in the haltere

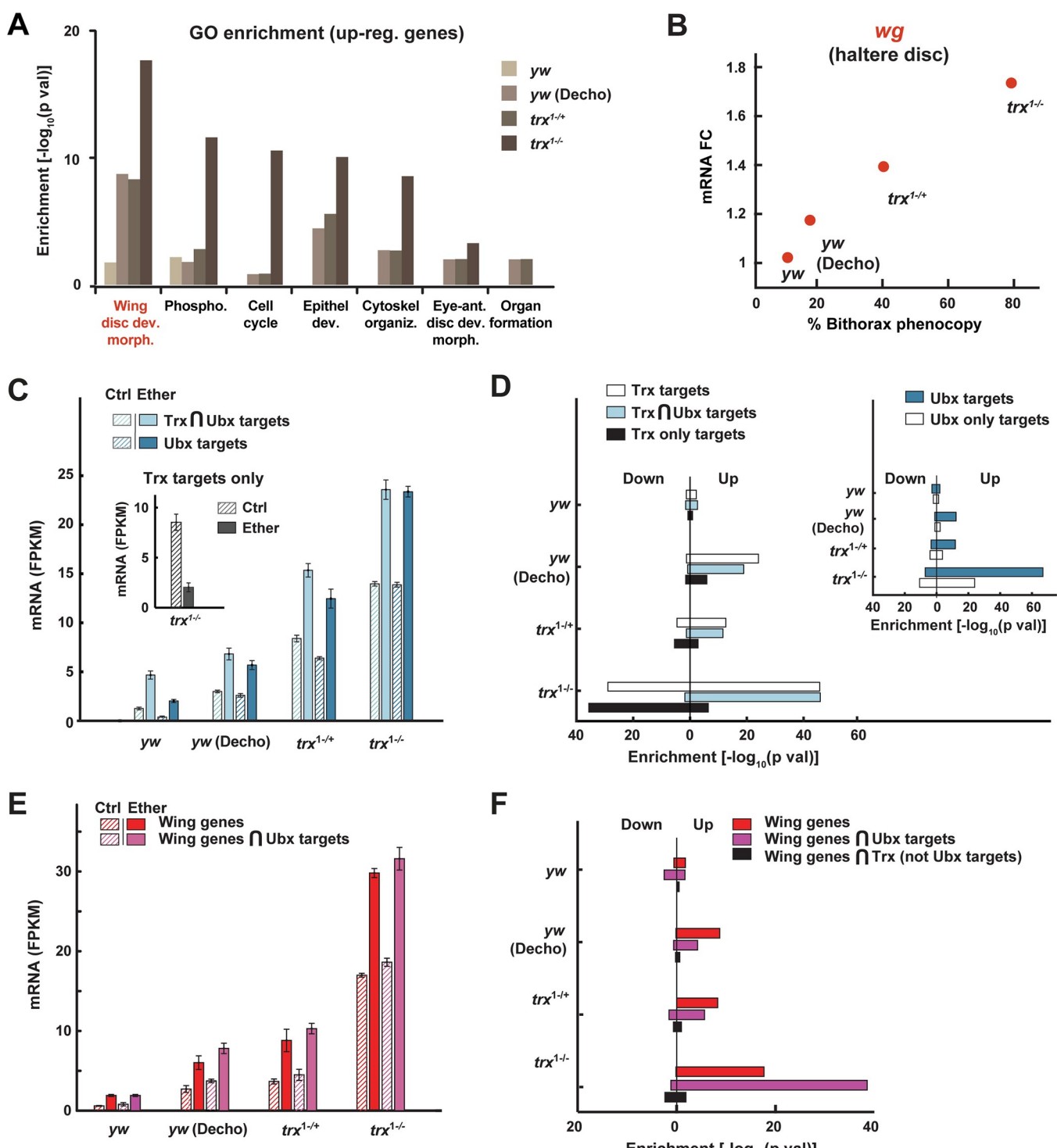

**Fig 4. Bithorax induction is manifested by up-regulation of wing genes and Ubx-Trx targets in the haltere.** (A) Functional enrichments of GO terms in genes that were significantly up-regulated by ether in haltere discs of third instar larvae from *trx*[1-/+] and *trx*[1-/-] stocks, and a wild-type (*yw*) stock, with and without egg dechorionation ("Decho") prior to exposure. Based on David online tool. (B) Ether-induced mRNA fold-change of wingless in the haltere discs. Mean fold-change (ether vs. control) for the cases in (A). (C) Median levels of mRNA ± SE for the indicated subsets of genes. *n* = 3. Ether effect: *p* < 0.001, Genotype effect: *p* < 0.001, Ether-Genotype interaction: p < 0.001, two-way ANOVA (full set of ANOVA *p*-values provided in S3 Table). (D) Enrichment of Trx and Ubx targets (inset) within up- and down-regulated genes in the haltere discs (over 1.5-fold up or down and *p* < 0.05), Fisher exact test. (E, F) Same as (C, D) for wing development genes and their intersection with Ubx targets or with Trx targets that are not shared with Ubx. Ether effect: *p* < 0.001, Genotype effect: *p* < 0.001, Ether-Genotype interaction: *p* < 0.001, two-way ANOVA (full set of ANOVA *p*-values provided in S4 Table). mRNA analyses are based on 3 biological replicates for each genotype. The data underlying this figure can be found in S3 Data.

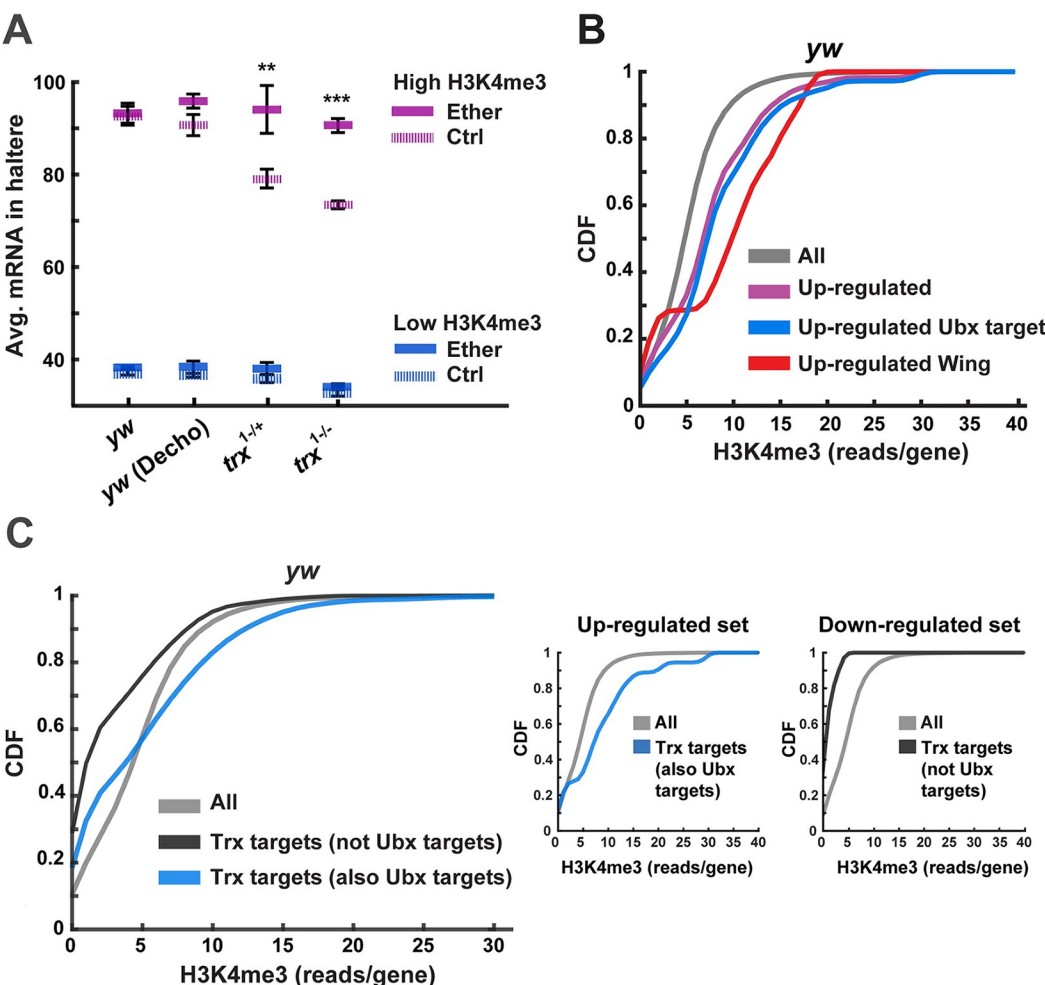

**Fig 5. Linkage between higher H3K4me3 levels of wing genes shortly after exposure and their up-regulation in the haltere. (A)** Transcriptional levels in the haltere (third instar larvae) for genes with the highest (top 10%, purple) and lowest (bottom 10%, blue) levels of H3K4me3 shortly after exposure to ether (4.5 h AED). Shown for *yw* (with and without dechorionation), *trx*$^{1-/+}$, and *trx*$^{1-/-}$. Mean FPKM ± SE, *n* = 3. **$^{**}p < 0.01$, $^{***}p < 0.001$, two-way ANOVA followed by Tukey HSD test. **(B)** Cumulative distribution function (CDF) of embryonic H3K4me3 levels shown for up-regulated genes (purple), up-regulated Ubx targets (blue), up-regulated wing development genes (red), and all genes (gray). **(C)** Same as (B) for joint targets of Trx and Ubx (blue), targets of Trx that are not shared with Ubx (black), and all genes with detectable methylation (gray). Inset: Same for the indicated subsets of targets that are up- and down-regulated in the haltere disc (left and right, respectively). The data underlying this figure can be found in S4 Data.

discs of third instar larvae. We found that genes with high and low H3K4me3 levels in the embryo (top and bottom 10%) are expressed, respectively, at high and low levels in the haltere (Fig 5A). Reciprocal analysis of genes that are up-regulated in the haltere showed that the embryonic H3K4me3 levels of these genes are higher compared to bulk (Figs 5B and S8A and S5 Table). The preferentially higher levels of H3K4me3 were even more pronounced in the subset of up-regulated wing genes versus the entire set of up-regulated genes (Fig 5B). The linkage between early H3K4me3 and later expression of wing genes in the haltere was further supported by differential methylation of distinct subsets of Trx targets (Fig 5C). In particular, the triimethylation of H3K4 of joint targets of Trx and Ubx was significantly shifted towards higher levels (Fig 5C, left inset; $p < 1E-5$), while the H3K4 trimethylation of Trx targets that are not shared with *Ubx* was shifted toward lower levels (Fig 5C, right inset; $p < 1E-5$). These

differences in H3K4me3 were consistent with matching (positive and negative) transcriptional fold changes in the haltere disc (insets to Figs 5C, S8B, and S8C). Taken together, these findings implicate wing genes-related differences in H3K4me3 shortly after exposure with later-stage induction of haltere-to-wing transformations.

## Discussion

Fossil record evidence suggests that the common ancestor of all winged insects had 2 pairs of large membranous flight wings, located on the second and third thoracic segments [54]. In flies and various orders of insects, the hindwings evolved into organs with altered functions, such as the gyroscopic haltere in *Drosophila*. These alterations appear to have emerged as co-options of a wing program, as evidenced by reversals to a double pair of wings on the background of mutations in genes of the bithorax gene complex, such as *Ubx* and *trx* [16,17,19,55]. Trx is the methylase responsible for deposition of the histone H3K4me3 mark that contributes to maintenance of an active state of expression of many genes, including the homeotic genes that specify segment identities in the *Drosophila* embryo (e.g., the Antennapedia complex and Bithorax complex genes) [47–49]. Ubx, in turn, is a Trx target and a master regulator of haltere development that specifies haltere fates in *Drosophila* by repressing the transcription of multiple wing genes in the third thoracic segment. Loss-of-function mutations in *Ubx* and *trx* are therefore consistent with spontaneous haltere-to-wing transformations. In a seminal work on gene–environment interactions, Waddington demonstrated that haltere-to-wing transformations can also be induced by embryonic exposure of wild-type embryos to ether vapor [9,10]. Similar induction was demonstrated by exposure to heat at the same time window of sensitivity [32,34,56], but the mechanistic basis of induction remained unknown for over 60 years. Our results show that brief exposure to ether at the time of cellularization (Fig 6) alleviates the suppression of Ubx-dependent wing genes in the larval haltere discs. This "reprogramming" of the larval disc is mirrored in ether-exposed embryos by higher levels of H3K4 trimethylation of wing genes versus bulk. This is, in turn, expected to assist in preferential maintenance of their active state of transcription over time [52], which is consistent with the haltere-to-wing predisposition. While investigating how early exposure to ether creates this predisposition, we discovered that ether vapor dissolves the eggshell and compromises protein integrity in the embryo. This increases the demand for Hsp90, a central chaperone that assists the folding of a wide range of proteins ("Hsp90 clients") during normal development, especially under stress. The chaperone activity of Hsp90 is required to support diverse functions, including transcriptional regulation, chromatin remodeling, and phenotypic buffering of genetic and epigenetic variations [21,22,57–65]. Since Hsp90 has also been implicated in the function of *Drosophila* Trx, as well as in the methyltransferase function of SMYD3 in mammals [33,66], we suspected that the proteotoxic stress in ether-exposed embryos compromises Trx function by altering the deployment of Hsp90. This was supported in this work by several lines of evidence, including enhanced induction of bithorax phenocopies on the background of genetic or chemical reduction in Hsp90 and the epistasis between reduced functions of *Hsp90* and *trx*. These findings implicate the environmental impact on protein integrity and chaperone load with an altered pattern of epigenetic memory that predisposes the haltere segment for partial reversal toward wing.

By integrating our findings with evidence from previous studies, we propose a model that accounts for bithorax-like transformations in response to early embryonic exposure to both ether and heat (Fig 6). Exposure at around the time of cellularization creates proteomic stress and subsequent redeployment of the *Drosophila* Hsp90 (Hsp83) towards misfolded proteins. The reduced availability of Hsp90 for Trx at a stage in which Trx function is particularly

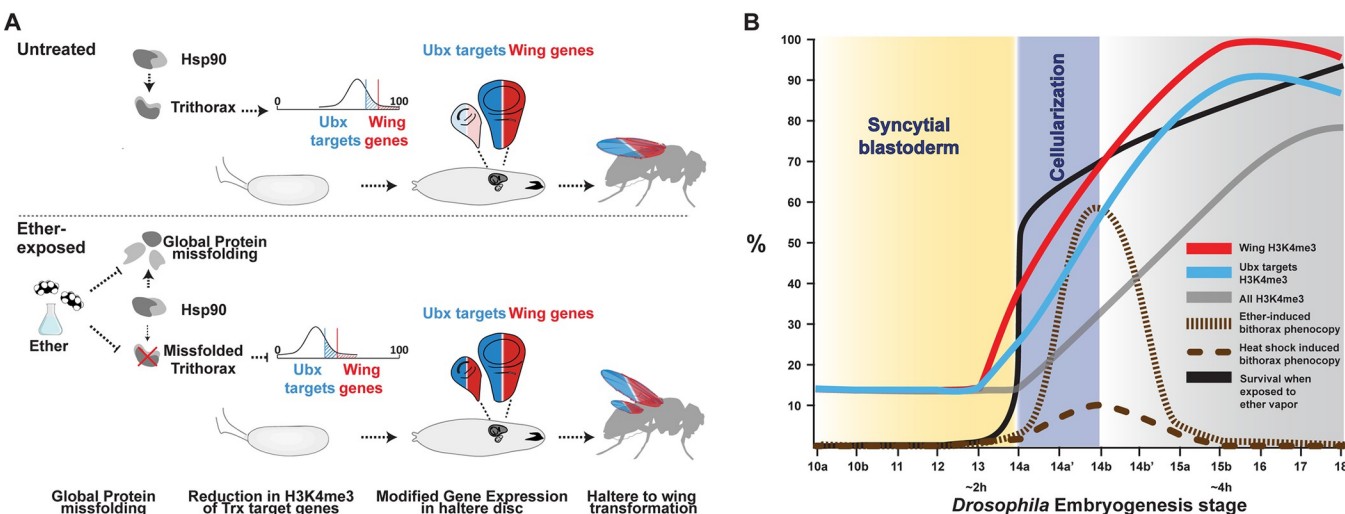

**Fig 6. Hypothetical model of induced bithorax phenocopies. (A)** Redeployment of Hsp90 towards misfolded proteins interferes with its contribution to Trx function and compromises the establishment of H3K4me3 marks. Genes that were more highly tri-methylated prior to the exposure and/or differentially retain H3K4me3 marks during the exposure, also exhibit higher H3K4me3 after the exposure. This set of genes is highly enriched with wing development genes that are normally repressed in the haltere by Ubx. Following the exposure, the relatively higher H3K4me3 levels of wing genes and Ubx targets (vs. bulk) contribute to differentially higher stability of their active transcriptional state during later stages of development. Active transcription of wing genes that are normally suppressed by Ubx in the haltere promotes ectopic induction of haltere-to-wing phenocopies, which can be viewed as partial reversals to the ancestral wing patterning. **(B)** Consistency of the proposed model with multiple lines of independent evidence. The lethality of exposure during most of the syncytial period [8,32,35] can be readily explained by higher permeability to the vapor (compared to exposure post-cellularization), thus aggravating the disruption of proteins that are critical for patterning the egg cytoplasm. The jump in survival for exposures after the onset of cellularization and the progressively increasing efficacy of induction between stages 14 and 15 are explained, respectively, by reduced permeability to the vapor and gradual priming of H3K4me3 in Ubx targets and wing development genes (modENCODE data [72]). The combined effects establish an intermediate "window of opportunity" for viable induction of bithorax transformations. Since Hsp90 is also required for mitigating the impacts of high temperature, the proposed model can also account for the reported induction of bithorax phenocopies by exposure to heat shock during this window [32,34]. Subsequent reduction in the efficacy of the bithorax induction after stage 15a [8,32,35] is consistent with substantial establishment of H3K4me3 marks by this stage [72]), as well as with the reduced ability of the vapor to penetrate the multicellular structures and tissue barriers. Developmental stages correspond to the nomenclature in [32].

required (Fig 6A and 6B [67,68]) interferes with the setup of H3K4me3. The expression of genes with relatively high post-exposure levels of H3K4me3 (e.g., wing development genes that are targeted by Ubx) is expected to be more stably sustained at a later stage of development. This is supported by the alleviated suppression of Ubx targets in the third thoracic segment (whose identity is specified by Ubx) and the induction of wing phenotypes in the haltere.

In addition to providing a plausible explanation for the induction of bithorax phenocopies by ether (or heat), the chain of events identified in this work can also account for the lethality of exposure at an earlier stage as well as the lack of responsiveness to exposure a few hours later (Fig 6B). Previous work showed that the induction of bithorax phenocopies by either ether or heat is largely confined to a time window that starts shortly before cellularization and ends at around the stage of partial invagination of the anterior and posterior midgut. Exposure during syncytial stages (<2 h AED) leads to complete embryonic lethality, while exposure after furrow formation (>4 h AED) is no longer capable of inducing a phenocopy. In between, the survival increases as a function of the onset of exposure, while the penetrance increases to a peak at around the end of cellularization and gradually decreases at later onsets of exposure (Fig 6B). This phenomenology is fully consistent with the hypothesized involvement of Hsp90 and Trx functions; early embryonic stages are characterized by rapid divisions of nuclei within a large cytoplasmic compartment. This cytoplasm is initially loaded with very high levels of maternal transcripts of Hsp90 (modENCODE data [69]), which contributes to protein folding and functional integrity in this large compartment. The activity of Hsp90 at this stage may be

particularly critical for maintaining the cytoplasmic protein gradients that specify the anterior-posterior and lateral-ventral axes [70]. Since the interruption of these gradients is lethal [71], a sufficient disruption of protein integrity can account for the lethality of exposure at that stage (Fig 6B). This was indeed supported by the 2 categories of defects that have been observed in the case of early exposure [35], namely: (i) failure to form a blastoderm, resulting in an undifferentiated-like mass with no recognizable structures; and (ii) emergence of anterior, posterior, or segmentation defects that are eventually followed by failure to hatch. As expected, these abnormalities are also more pronounced in embryos that were exposed at progressively earlier stages [35].

Altogether, these findings portray a causal chain of events, connecting environmental disruption of protein integrity at the onset of histone methylations with modified epigenetic patterning that supports a morphogenetic shift towards an ancestral-like body plan. The increased morphogenetic sensitivity to the reduction of Hsp90 and Trx functions is likely to be involved in other contexts of induced homeotic transformations [16,21] and may be utilized for fate manipulation and/or regeneration at the levels of tissues and organs.

## Materials and methods

### *Drosophila* stocks

*Drosophila* lines $trx^1$/TM1 and $Hsp83^{e6A}$ were obtained from the Bloomington Stock Center. The *yw* stock was obtained from the laboratory of Prof. Eli Arama (Weizmann Institute of Science, Israel).

### Generation of $trx^1$/TM6B, Tb flies

We have replaced the third chromosome balancer of the original line $trx^1$/TM1 with a Tm6b balancer carrying a larval marker (Tb) to have the ability to differentiate between homozygous and heterozygous larvae. We have restored the genetic background of the original line [16], excluding the Tm6b, using chromosomal markers.

### Food preparation

Standard cornmeal food (Bloomington Stock Center recipe, http://flystocks.bio.indiana.edu/Fly_Work/media-recipes/molassesfood.htm).

### Exposure and scoring of responses to ether

The 0- to 3-day-old flies were reared on standard fly medium for 3 days under a 12 h light/dark cycle regime, temperature of 25°C, and 70% humidity. About 10,000 adult flies were taken for 5 rounds of egg deposition for 1 h in cages placed on 10 cm agar plates. The first 2 rounds of egg deposition were discarded for synchronization of the embryos' developmental stage. Dechorionation was performed 2 h later by exposing the eggs to 3% sodium hypochlorite solution for 2.5 min and washing with tap water. Embryos at the syncytial blastoderm stage 2.5 to 3.5 h after oviposition [8] were placed in a 100 μm Cell Strainer (Falcon) and transferred to a glass bottle containing 1% agarose. Eppendorf tube containing 1.5 ml of diethyl ether was added to the glass bottle, and the lid closed, allowing exposure of embryos to the vapor for 30 min at 25°C. After 30 min, the tube was removed. For embryonic RNA, protein, and ChIP analyses, 3:45 to 4:45 h old embryos were collected and flash-frozen in liquid nitrogen. For haltere disc RNA analysis and phenocopy scoring, embryos were transferred to new bottles with fly medium and allowed to hatch. Haltere discs were dissected from third instar larvae developed from ether exposed and non-exposed embryos. Phenocopy scoring was

performed daily, starting from the first day of eclosion and continuing for 5 days. Following day 5, all the unenclosed pupae were dissected and scored for phenocopies. For Hsp90 inhibition experiments, embryos were dechorionated and rocked in PBS supplemented with 35 μm Geldanamycin (Sigma-Aldrich) for 1 h [46] prior to the ether exposure procedure.

## Chromatin immunoprecipitation (ChIP) and ChIP-seq

Ether exposed and non-exposed embryos (3:45 to 4:45 h old) were collected (0.1 mg per group). Embryos were crosslinked in 1 ml A1 buffer (60 mM KCl, 15 mM NaCl, 15 mM HEPES [pH 7.6], 4 mM MgCl2, 0.5% Triton X-100, 0.5 mM dithiothreitol (DTT), and complete EDTA-free protease inhibitor cocktail [Roche]), in the presence of 1.8% formaldehyde and homogenized at the same time in a douncer, followed by incubation for 15 min at room temperature. Crosslinking was stopped by adding 225 mM glycine, followed by incubation for 5 min. The homogenate was transferred to a 1 ml tube and centrifuged for 5 min, 4,000 × g at 4˚C. The supernatant was discarded, and the nuclear pellet was washed 3 times in 3 ml A1 buffer and once in 3 ml of A2 buffer (140 mM NaCl, 15 mM HEPES [pH 7.6], 1 mM EDTA, 0.5mMEGTA, 1%Triton X-100, 0.5mMDTT, 0.1% sodium deoxycholate, and protease inhibitors) at 4˚C. After the washes, nuclei were resuspended in A2 buffer in the presence of 0.1% SDS and 0.5% N-lauroylsarcosine and incubated for 30 min on a rotating wheel at 4˚C. Chromatin was sonicated using a Bioruptor (Diagenode) for 15 min (settings 30 s on, 30 s off, high power). Sheared chromatin had an average length of 300 to 700 base pairs. After sonication and 10-min high-speed centrifugation, fragmented chromatin was recovered in the supernatant. Chromatin was precleared by the addition of 50 μl of Protein A-Agarose (PA) suspension (Roche 11134515001) followed by overnight incubation at 4˚C. PA was removed by centrifugation, antibodies at dilution 1:100 were added to the supernatant (control in the presence of rabbit pre-serum [Mock IP] was performed at the same time), and samples were incubated for 4 h at 4˚C in a rotating wheel. PA (50 μl) was added, and incubation was continued overnight at 4˚C. Antibody-protein complexes were collected by centrifugation at 4,000 rpm for 1 min, and the supernatants were discarded. Samples were washed 4 times in A3 (A2+ 0.05% SDS) buffer and twice in 1 mM EDTA, 10 mM Tris (pH 8) buffer (each wash, 5 min at 4˚C). Chromatin was eluted from PA in 250 μl of 10 mM EDTA, 1% SDS, 50 mM Tris (pH 8) at 65˚C for 15 min, followed by centrifugation and recovery of the supernatant. The eluate was incubated overnight at 65˚C to reverse crosslinks and treated with Proteinase K for 3 h at 50˚C. Sodium acetate (110 μm) was added to the samples, phenol–chloroform extraction, and ethanol precipitation in the presence of 20 μg glycogen. DNA was resuspended in 100 μl of water. Deep sequencing analysis of DNA was performed by Fasteris SA (Geneva, Switzerland). ChIP-seq library preparation was performed with Illumina TruSeq ChIP kit. Adaptors were removed from raw FASTQ reads with cutadapt. The reads were then aligned to the *Drosophila* genome (UCSC dm3) with bowtie2; samtools and bedtools were used to convert resulting SAM files to the required downstream formats (bedgraph etc.). We performed the analysis via the "Misha" R package [51]. The signal was smoothed via a moving window averaging over 100 bp followed by global percentile normalization. Next, a 95% threshold was applied to separate the signal from the background.

## RNA-seq library preparation and sequencing

The cDNA libraries were prepared from poly-A mRNA following the manufacturer's instructions in the Illumina RNA sample preparation kit. In short, poly (T) oligo-attached magnetic beads were used to purify the poly(A)-containing mRNA molecules. The mRNA was fragmented into 200 to 500 bp segments. RNA fragments were converted into cDNA using

SuperScript II reverse transcriptase (Life Technology) and random hexamer primers. Adaptors were ligated to the cDNA fragments, followed by purification, PCR, and additional purification. Deep sequencing measurement of RNA was performed in the Genomics Core Facility unit of the Technion Genome Center, Technion—Israel Institute of Technology (Haifa, Israel) using Illumina Genome Analyzer IIx (GA IIx). For sequencing, we used the following experimental kits and reagents: (i) Standard Cluster Generation Kit (#GD-103-4001, Illumina, San Diego, California, United States of America) containing all reagents necessary to load the samples onto the flow cell and perform the bridge amplification; (ii) Illumina Sequencing Kit v5 (TruSeq SBS Kit v5 GA (36-cycles), FC-104-5001), which contains the reagents for the sequencing runs; and (iii) the GA IIx Sequencing Control Software version SCS 2.8, which was used to control the sequencer. Sequencing was based on 50 bp single-end reads. mRNA was barcoded in the ligation step by Illumina standard multiplex adaptors. The multiplexed samples were sequenced on a single lane to yield between 2 and 8 million reads per sample.

## RNA-seq analysis

Adaptors were removed from sequence reads using the cutadapt program [73]. Reads were mapped to the *Drosophila* transcriptome (Ensembl version BDGP.25) using Bowtie2 and TopHat software, then Cufflinks and Cuffmerge [74] were applied to define a list of transcripts that are comparable between all samples. Differentially expressed transcripts, including fold-change and statistics, were identified by applying the DESeq R package [75] on the bowtie2 output. GO enrichments were computed using the "DAVID" online resource with cutoffs for up-/down-regulation and FDR set to 1.5-fold and 0.05, respectively. Up- and down-regulated gene sets were analyzed separately. Targets of Trx and Ubx in the haltere were derived from [76–78].

## Eggshell permeabilization

Several hundred *yw* adult flies were synchronized twice for 1 h and allowed to lay eggs for 1 h on a 10 cm agar plate. Eggs were collected from the plate, washed in water, dechorionated, and immersed in Citrasolv (Citra Solv, Danbury, Connecticut) (1:10 dilution, 5 min), diethyl ether (5 min), or were exposed to diethyl ether vapors for 1.5 h in a closed bottle. Control embryos were left untreated in a closed bottle for the same period of time. Then, the embryos were stained with Acridin orange dye for 5 min. Images were taken using a fluorescent stereoscope LEICA MZ16F equipped with a Nikon digital sight DS-Fi1 camera.

## Circular dichroism

CD spectra were recorded on a Chirascan spectropolarimeter (Applied Photophysics) calibrated with a solution of ammonium d-10-camphorsulfonate. Far-UV CD spectra were acquired using 1-mm path-length cuvettes, a step size of 0.5 nm, a bandwidth of 1 nm, and a time constant of 1 s. Protein concentration was 0.4 mg/ml. Following the acquisition, the experiment was corrected for buffer contributions, averaged, and smoothed using sliding windows of 1.5 nm, far- and near-UV.

## Fluorescence spectroscopy

*D. melanogaster* embryos were collected and exposed to ether as described in "Exposure and scoring of responses to ether." For heat exposure, embryos were incubated in Eppendorf tubes at 80°C. Control, ether-treated, and heat-treated embryos were lysed and homogenized in 10 mM phosphate buffer using a "loose" pestle with 6 to 8 strokes. The homogenate was

centrifuged at 12,000 g for 5 min to pellet nuclei, and the supernatant was transferred to a new tube. RNase (2 μl of 0.5 mg/ml) was added, followed by incubation at 25°C for 1 h. Protein samples (0.1 to 0.4 mg/ml) were then incubated with 0.2 mM 8-anilino-1-naphthalenesulfonate (ANS) (Sigma-Aldrich) for 1 h. Fluorescence was measured using a Cytation 5 plate reader (BioTek) with excitation at 380 nm and emission at 470 nm for the dye-protein complex and 545 nm for unbound dye. Data were corrected for dye contributions, averaged, and smoothed using 3 nm sliding windows.

## Quantification of RFP fluorescence

Several hundred His3Av-mRFP1 adult flies were synchronized twice for 1 h and allowed to lay eggs for 1 h on a 10 cm agar plate. Eggs were collected from the plate, washed in water, dechorionated, and exposed to ether as previously described. Eggs were then transferred to MatTek Glass-Bottom Dishes that were pretreated with embryo glue (3M tape in Heptane) and covered in Halocarbon oil 700 (Sigma). The eggs were imaged using UPLSAPO 20× numerical aperture: 0.75 objective of the confocal OLYMPUS FV1000 microscope with temperature-controlled chamber (set at 25°C) and IX81 ZDC Motorized Stage. Image analysis was performed using ImageJ.

## Proteomics

**Sample preparation.** *D. melanogaster* embryos were collected and exposed to ether as described in "Exposure and scoring of responses to ether." Control and ether-treated embryos (0.1 mg per group) were lysed and homogenized in 50 mM Tris-HCl with 5% SDS. Lysates were incubated at 96°C for 5 min, followed by 6 cycles of 30-s sonication (Bioruptor Pico, Diagenode, USA). Protein concentration was measured using the BCA assay (Thermo Scientific, USA). A total of 100 μg protein was reduced by treatment with 5 mM dithiothreitol and alkylated with 10 mM iodoacetamide in the dark. Each sample was loaded onto S-Trap mini-columns (Protifi, USA) according to the manufacturer's instructions. In brief, loaded samples were washed with 90:10% methanol/50 mM ammonium bicarbonate, digested with trypsin (1:50 trypsin/protein) for 1.5 h at 47°C. The digested peptides were eluted using 50 mM ammonium bicarbonate and incubated overnight with trypsin at 37°C. Two additional elutions were made using 0.2% formic acid and 0.2% formic acid in 50% acetonitrile. The 3 elutions were pooled together and vacuum-centrifuged to dry and samples were kept at −80°C until the analysis.

**Liquid chromatography.** ULC/MS grade solvents were used for all chromatographic steps. Each sample was fractionated using high pH reversed phase followed by low pH reversed phase separation, and 100 μg digested protein was loaded using high Performance Liquid Chromatography (Acquity H Class Bio, Waters, Milford, Massachusetts, USA). Mobile phase was: (i) 20 mM ammonium formate pH 10.0; (ii) acetonitrile. Peptides were separated on an XBridge C18 column (3 × 100 mm, Waters) using the following gradient: 3% B for 2 min, linear gradient to 40% B in 50 min, 5 min to 95% B, maintained at 95% B for 5 min and then back to initial conditions. Peptides were fractionated into 15 fractions. The fractions were then pooled: first fraction: 1 with 2, 3, 13, 14, and 15, second fraction: 4 with 9, third fraction: 5 with 10, fourth fraction: 6 with 11, fifth fraction: 8 with 12, sixth fraction: 7 only. Each fraction was vacuum dried, then reconstituted in 50 μl 97:3 acetonitrile/water + 0.1% formic acid. Each pooled fraction was loaded using split-less nano-Ultra Performance Liquid Chromatography (10 kpsi nanoAcquity, Waters, Milford, Massachusetts, USA). The mobile phase was: (i) water + 0.1% formic acid and (ii) acetonitrile + 0.1% formic acid. Desalting of the samples was performed online using a reversed-phase Symmetry C18 trapping column (180 μm internal

diameter, 20 mm length, 5 μm particle size; Waters). The peptides were then separated using a T3 HSS nano-column (75 μm internal diameter, 250 mm length, 1.8 μm particle size; Waters) at 0.35 μl/min. Peptides were eluted from the column into the mass spectrometer using the following gradient: 4% to 25% B in 105 min, 25% to 90% B in 5 min, maintained at 90% for 5 min and then back to initial conditions.

**Mass spectrometry.** The nanoUPLC was coupled online through a nanoESI emitter (10 μm tip; New Objective; Woburn, Massachusetts, USA) to a quadrupole orbitrap mass spectrometer (Q Exactive HF, Thermo Scientific) using a FlexIon nanospray apparatus (Proxeon). Data was acquired in data-dependent acquisition (DDA) mode, using a Top20 method. MS1 resolution was set to 120,000 (at 400 m/z), mass range of 375 to 1,650 m/z, AGC of 3e6 and maximum injection time was set to 50 msec. MS2 resolution was set to 15,000, quadrupole isolation 1.7 m/z, AGC of 1e5, dynamic exclusion of 40 s, and maximum injection time of 60 msec. A preferential inclusion list was specified for higher priority of MS/MS triggering.

**Data processing.** Raw data was processed with MaxQuant v1.6.0.16. The data was searched with the Andromeda search engine against the *Drosophila melanogaster* proteome database (March 2018 version) appended with common lab protein contaminants and the following modifications: Carbamidomethylation of C as a fixed modification and oxidation of M, deamidation of N and Q and protein N-terminal acetylation as variable ones. The default parameters were used with the following changes: Minimal LFQ peptide ratio count was set to 1, match between runs option enabled, as well as the iBAQ calculation. The LFQ (Label-Free Quantification) intensities were calculated and used for further calculations using Perseus v1.6.0.7. Decoy hits were filtered out, as well as proteins that were identified on the basis of a modified peptide only. GO annotations were added and the common contaminates are labeled with a "+" sign in the relevant column. The LFQ intensities were used to calculate the ratio for each protein between the different samples. Embryonic targets of Hsp90, Trx, and Ubx were derived from [78–81].

## Western blotting

*D. melanogaster* flies (5 to 8 day old) were transferred into embryo collection cages with 10 cm agar plates supplemented with yeast paste. The first 2 rounds of 1-min anesthesia, followed by 1-h oviposition were used for synchronization of the embryos' developmental stage. The embryos from the third egg-laying round were collected, dechorionated, and exposed to ether as described in "Exposure and scoring of responses to ether." Control and ether-treated embryos were then lysed with RIPA buffer containing protein inhibitor cocktail (ApexBio #K1007). Clarified *D. melanogaster* embryo lysates were concentrated (2.5-fold) using 3 kDa MWCO Vivaspin 500 centrifugal concentrators (Sartorious) according to the manufacturer's protocol. Total protein concentrations were determined by a Bradford assay (Sigma-Aldrich) using a BSA protein standard calibration curve. Samples (220 mg total protein per lane) were loaded in duplicates onto reducing 4% to 20% ExpressPlus precast PAGE gels (GenScript), alongside an MW marker (PageRuler Plus prestained protein ladder, 10 to 250 kDa, Thermo Fisher Scientific). For visualization, gels were stained using InstantBlue Coomassie stain (expedeon). For western blotting, gels were transferred onto nitrocellulose membranes 0.45 μm (Protran, Whatman) using a Mini Trans-BlotCell transfer system (Bio-Rad) and blocked with 5% skim milk powder in PBS with 0.5% Tween-20 (RT, 1 h). The Ubx protein was detected using mouse anti-Ubx primary antibody (1:100 dilution, 4˚C, O/N, DSHB, University of Iowa, Iowa, catalog #Ubx FP3.380s) and HRP-conjugated goat anti-mouse IgG secondary antibody (1:5,000 dilution, RT, 1 h, Jackson ImmunoResearch, catalog ##115-035-062). α-tubulin was detected using a mouse anti-α-tubulin primary antibody (1:10,000 dilution, 1 h, RT, DSHB,

University of Iowa, Iowa, catalog #12G10 anti-α tubulin-s) and HRP-conjugated goat anti-mouse IgG secondary antibody (1:10,000 dilution, RT, 1 h). Primary and secondary antibodies were diluted in 2% skim milk powder in PBS with 0.5% Tween-20. Membranes were imaged using the IQ800 imaging system (Cytiva) and band densities were quantified using Image-Quant TL analysis software. The Hsp83 protein was detected using rabbit anti-Hsp83 primary antibody (1:480 dilution, 4°C, O/N, DSHB, obtained from Paro lab (Tariq and colleagues) and HRP-conjugated goat anti-rabbit IgG secondary antibody (1:20,000 dilution, RT, 1 h, Jackson ImmunoResearch, catalog ##115-035-062).

## Reverse transcription and quantitative PCR (qPCR)

RNA was extracted from 7- to 8-h-old embryos at 4 h after ether exposure using Quick RNA microPep (Zymo Research). cDNA was synthesized by a high-capacity reverse transcription kit (Thermo Fisher Scientific). Expression levels of representative genes, involved in wing development (*wg*, *bs*, *vg*, *ac*, *salr*, *ash2*, *exd*, *hth*, *ast*) and house-keeping gene (*act5C*) were analyzed by qPCR with fast SYBR green master mix (Applied Biosystems) on QuantStudio Absolute Q Digital PCR System (Applied Biosystems), and 5 ng of cDNA were used per sample with 3 technical replicates. Primer sequences, listed in S6 Table, were designed using FlyPrimerBank (http://www.flyrnai.org/flyprimerbank), except for *Act5C* [82,83]. The 2ΔΔCt method was used to analyze gene expression after normalization to *Act5C*.

## Statistical analyses

Statistical tests were performed using MATLAB (MathWorks) and R statistical program [84]. Significant differences between a subset group of genes to the entire population in their H3K4me3 or mRNA levels were numerically calculated by using a bootstrap-based statistical test, as follows: this test was based on repeated cycles (1,000,000) of selecting genes at random from the total set of genes (same sample size as the subset group) and counting the fraction of times, in which the median methylation/expression in this random selection exceeds or fell behind the median level of the true subset. Significance was determined based on the percentage of iterations in which this analytical *p*-value was equal to, lower, or higher than the median of the total set of genes. Analysis of enrichment of gene ontology annotations in sets of up- and down-regulated genes was done using the DAVID web tool with Benjamini correction for multiple hypotheses testing [85,86].

## Supporting information

**S1 Fig. Western blot analysis of Hsp83 protein levels in ether-treated and untreated embryos. (A)** The amount of Hsp83 protein was evaluated in 3 biological replicates of lysates of untreated eggs (C1-3) and lysates of ether-treated eggs (E1-3). Blots were initially probed using anti-Hsp83 and then with anti-α-Tubulin antibodies. **(B)** A Coomassie-stained SDS-PAGE gel of identical samples prior to blotting is shown for comparison. M, Molecular weight marker.
(AI)

**S2 Fig. Western blot analysis of Ubx protein levels in ether-treated and untreated embryos.** The amount of Ubx protein was evaluated in 3 biological replicates of lysates of untreated eggs (C1-3) and ether-treated eggs (E1-3). **(A, B)** Blots were initially probed using anti-Ubx (A) and then with anti-α-Tubulin (B) antibodies. **(C)** A Coomassie-stained SDS-PAGE gel of identical samples prior to blotting is shown for comparison. M, Molecular weight marker.
(AI)

**S3 Fig. Representative ChIP-seq profiles.** H3K4me3 (left) and H3K27me3 (right) profiles are shown for the indicated genes in ether-exposed (red) and non-exposed embryos (blue). Vertical and horizontal arrows indicate reduction in the number of H3K4me3 reads and a lack of significant change in the H3K27me3 reads, respectively.
(AI)

**S4 Fig.** **(A)** Normalized numbers of genomic regions with H3K4 and H3K27 tri-methylation (100 bp and 1,000 bp long, respectively), displayed for ether-exposed and control embryos (*yw*). **(B–D)** mRNA levels in *yw* and *trx^1^* stocks in ether-exposed (y-axis) vs. non-exposed control embryos (x-axis). Differential expression (absolute fold-change >1.5, $p < 0.05$, $n = 3$) is indicated by red and purple overlays. **(E)** mRNA fold-change of wing genes in ether-exposed and control embryos, analyzed by qPCR 4 h after exposure (8–9 h AED). Shown for a panel of genes that are known to be involved in wing formation and are also up-regulated in the haltere discs of larvae that were developed from ether-exposed embryos. **(F)** Intersection between wing disc developmental genes and genes with highest and lowest H3K4me3 levels (top and bottom 10%) shortly after exposure to ether (top) or no exposure (bottom). *** $p < 1E-8$, *** $p < 1E-14$, respectively, hypergeometric test. **(G)** Box plots of H3K4me3 read counts corresponding to the 0–4 h and 4–8 h AED time windows of embryonic development. Displayed for all genes and wing development genes ("Wing") based on compilation of ModEncode data [19]. The data underlying this figure can be found in S5 Data.
(AI)

**S5 Fig. Ether suppresses H3K4me3 tri-methylation, primarily in low-expressed loci.** **(A)** Intersection between genes with high preferential retention of H3K4me3 marks ("Most retained," blue) and genes with highest and lowest expression in control embryos. *** $p < 1E-7$. Same for the intersection of "Least retained" genes (purple). *** $p < 1E-22$. **(B)** Cumulative distribution function (CDF) of normalized H3K4me3 level per gene, shown for all genes with a detectable level of H3K4me3 (gray) and genes within the sets exhibiting 10% highest and lowest retention of H3K4me3 (blue and purple, respectively). *** $p < 1E-6$. **(C)** Intersection between genes with high preferential retention of H3K4me3 marks ("Most retained," blue) and genes with the highest and lowest H3K4me3 levels in control embryos (top and bottom 10%). *** $p < 1E-19$, hypergeometric test. Same for the intersection of "Least retained" genes (purple). *** $p < 1E-137$. **(D)** Median mRNA (solid line) and H3K4me3 levels (dashed line) for genes with high, medium, and no preferential retention. The data underlying this figure can be found in S6 Data.
(AI)

**S6 Fig.** **(A)** Number of differentially expressed genes (absolute fold-change >1.5, $p < 0.05$, $n = 3$) in ether-exposed *yw*, *yw* decho, *trx^1+/-^* and *trx^1-/-^* haltere imaginal discs. **(B)** *Ubx* mRNA in ether-exposed *yw*, *yw* decho, *trx^1+/-^* and *trx^1-/-^* haltere imaginal discs. Average ± SE, $n = 3$. Two-way ANOVA following Tukey HSD test. **(C)** Same as (B) for *trx* mRNA. The data underlying this figure can be found in S7 Data.
(AI)

**S7 Fig. Western blot analysis of Ubx protein in haltere imaginal discs.** **(A, B)** Evaluation of Ubx protein relative levels in three biological replicates of dissected halteres, with and without embryonic exposure to ether (E1-3 vs. C1-3, respectively). Blots were initially probed using anti-Ubx (A) and then with anti-αTubulin (B) antibodies. M, Molecular weight marker. **(C)** Quantitative analysis of the Ubx protein levels in (A), normalized to α-tubulin (B). Mean ± SE, $n = 4$. $p > 0.05$, Student's *t* test. The data underlying this figure can be found in S8 Data.
(AI)

**S8 Fig. Up-regulation of wing genes in the haltere correlates with high H3K4me3 at the time of exposure. (A)** Cumulative distribution function (CDF) of H3K4me3 levels shortly after ether exposure, shown for the following groups of genes: all genes with detectable H3K4me3 (gray), genes that are significantly up-regulated in the haltere disc (pink), up-regulated wing genes, and up-regulated Ubx targets (red and blue, respectively). Shown for dechorionated *yw* flies (left), *trx*$^{1+/-}$ flies (center), and *trx*$^{1-/-}$ (right). CDFs are based on the sum of normalized H3K4me3 reads per gene. **(B)** Same as (A) with up-regulated genes that are jointly targeted by trx and Ubx (blue) instead of last 2 groups of (A). **(C)** Same as (B) for the subset of genes that are significantly down-regulated in the haltere disc (pink) and down-regulated genes that are joint targets of trx and Ubx (black).
(AI)

**S1 Spreadsheet. mRNA levels (FPKM values) in ether-exposed and non-exposed *yw*, *yw* decho, *trx*$^{1+/-}$ and *trx*$^{1-/-}$ embryos.**
(XLSX)

**S2 Spreadsheet. mRNA levels (FPKM values) in haltere discs of third instar larvae (*yw*, *yw* decho, *trx*$^{1+/-}$ and *trx*$^{1-/-}$), with and without embryonic exposure to ether.**
(XLSX)

**S1 Table. Significance of gene–environment interaction effect on penetrance.** Based on the fraction of pupae and adults presenting bithorax phenocopies. Environmental conditions: Ether, Dechorionation ("Decho"), and no treatment control. Genotypes: *yw*, *trx*$^{1+/-}$, and *trx*$^{1-/-}$. Paired Tukey HSD analysis was applied to each of the indicated pairs.
(DOCX)

**S2 Table. Gene network enrichment analysis of the genes with promoter regions within the highest H3K4me3 retention levels (top 10%).**
(DOCX)

**S3 Table. ANOVA analysis of the effects of the indicated factors on mean expression of subsets of trx and Ubx targets in haltere discs of third instar larvae.**
(DOCX)

**S4 Table. ANOVA analysis of the effects of the indicated factors on mean expression of wing-related targets of Ubx in haltere discs of third instar larvae.**
(DOCX)

**S5 Table. Significance of overlap between H3K4me3 levels in exposed embryos (4.5 h AED) and differential expression (ether vs. control) in the haltere discs of third instar larvae. Shown for *yw* (with and without dechorionation), *trx*$^{1-/+}$, and *trx*$^{1-/-}$. Hypergeometric test.**
(DOCX)

**S6 Table. Primer sequences for qPCR analysis of mRNA levels of wing genes.**
(DOCX)

**S1 Data. Supporting information underlying Fig 1.**
(XLSX)

**S2 Data. Supporting information underlying Fig 2.**
(XLSX)

**S3 Data. Supporting information underlying Fig 4.**
(XLSX)

**S4 Data. Supporting information underlying Fig 5.**
(XLSX)

**S5 Data. Supporting information underlying S4 Fig.**
(XLSX)

**S6 Data. Supporting information underlying S5 Fig.**
(XLSX)

**S7 Data. Supporting information underlying S6 Fig.**
(XLSX)

**S8 Data. Supporting information underlying Fig S7.**
(XLSX)

**S1 Raw Images. Western blots raw images.**
(PDF)

## Acknowledgments

We thank Dr. Maor Knafo for helpful discussions and Dr. Meital Kupervaser from the De Botton Protein Profiling Institute of the Nancy and Stephen Grand Israel National Center for Personalized Medicine, Weizmann Institute of Science, for performing mass spectrometry proteomics.

## Author Contributions

**Conceptualization:** Orli Snir, Yoav Soen.

**Data curation:** Orli Snir.

**Formal analysis:** Orli Snir, Michael Elgart, Yulia Gnainsky, Moshe Goldsmith, Shlomi Dagan, Yoav Soen.

**Funding acquisition:** Yoav Soen.

**Investigation:** Orli Snir, Michael Elgart, Yulia Gnainsky, Moshe Goldsmith, Filippo Ciabrelli, Shlomi Dagan, Iris Aviezer, Elizabeth Stoops.

**Methodology:** Orli Snir, Michael Elgart, Giacomo Cavalli, Yoav Soen.

**Project administration:** Orli Snir.

**Resources:** Giacomo Cavalli, Yoav Soen.

**Supervision:** Yoav Soen.

**Visualization:** Orli Snir.

**Writing – original draft:** Orli Snir, Yoav Soen.

**Writing – review & editing:** Orli Snir, Michael Elgart, Filippo Ciabrelli, Giacomo Cavalli, Yoav Soen.

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
