## [Editor Report · Decision Letter 0]

24 Aug 2023

Dear Dr Snir, 

Thank you for submitting your manuscript entitled "Waddington revisited: organ transformation by environmental disruption of epigenetic memory" for consideration as a Research Article by PLOS Biology.

Your manuscript has now been evaluated by the PLOS Biology editorial staff as well as by an academic editor with relevant expertise and I am writing to let you know that we would like to send your submission out for external peer review.

Once your full submission is complete, your paper will undergo a series of checks in preparation for peer review. After your manuscript has passed the checks it will be sent out for review. To provide the metadata for your submission, please Login to Editorial Manager (https://www.editorialmanager.com/pbiology) within two working days, i.e. by Aug 28 2023 11:59PM.

Kind regards,

Lucas

Lucas Smith, Ph.D.

Senior Editor

PLOS Biology

lsmith@plos.org

---

## [Decision Letter · Decision Letter 1]

30 Oct 2023

Dear Dr Snir,

Thank you again for your patience while your manuscript "Waddington revisited: organ transformation by environmental disruption of epigenetic memory" was peer-reviewed at PLOS Biology. Your study has now been evaluated by the PLOS Biology editors, an Academic Editor with relevant expertise, and by several independent reviewers. 

In light of the reviews, which you will find at the end of this email, we would like to invite you to revise the work to thoroughly address the reviewers' reports.

As you will see below, the reviewers highlight that the study is interesting and generally well done, but they have a number of suggestions to improve the manuscript before publication and we think these should be carefully addressed. Additionally, we also have a number of editorial and policy related requests that will need to be addressed before publication. I have appended those below my signature as well. 

Given the extent of revision needed, we cannot make a decision about publication until we have seen the revised manuscript and your response to the reviewers' comments. Your revised manuscript may be sent for further evaluation by all or a subset of the reviewers.

**IMPORTANT - SUBMITTING YOUR REVISION**

*Re-submission Checklist*

*Published Peer Review*

Sincerely,

Lucas

Lucas Smith, Ph.D.

Senior Editor

PLOS Biology

lsmith@plos.org

EDITORIAL REQUESTS: 

1) INTRODUCTION/TITLE: After some discussion within the team, we would suggest that the framing of the introduction and title of your piece be broadened a bit. We do appreciate that part of the interest of the study is that you have elucidated epigenetic pathways underlying classic Waddintonian experiments - but with our broad readership in mind, we think it might be helpful to expand the intro and and adjust the title to do a slightly better job of explaining why, beyond this historical backdrop, this is still an important study. We would not ask you to remove the discussion of Waddinton completely, but would suggest you consider framing the study more broadly around the demonstration of a mechanistic link between environmental perturbations and long-lasting changes in phenotype via epigenetic changes.

Along those lines, we suggest the title be edited to make it more broadly accessible. For example, if you agree, we suggest something like: "Environmental disruption of protein integrity during embryonic development disrupts epigenetic memory in Drosophila"

2) ABSTRACT: Please note that per journal policy, the model system/species studied should be clearly stated in the abstract of your manuscript. 

3) BLURB: In the relevant section of our online system, please provide a blurb which (if accepted) will be included in our weekly and monthly Electronic Table of Contents, sent out to readers of PLOS Biology, and may be used to promote your article in social media. The blurb should be about 30-40 words long and is subject to editorial changes. It should, without exaggeration, entice people to read your manuscript. It should not be redundant with the title and should not contain acronyms or abbreviations.

4) DATA AVAILABILITY: We see that you have put data related to your paper on ProteomeXchange and SRA databases. Can you please provide us with accession numbers and reviewer tokens so that we can access that data?

Please do read our data policy, which requires that all data be made available without restriction, and make sure to provide the relevant underlying data for your study. http://journals.plos.org/plosbiology/s/data-availability

As detailed in the policy, we do not require all raw data. Rather, we ask that all individual quantitative observations that underlie the data summarized in the figures and results of your paper be made available in one of the following forms:

a. Supplementary files (e.g., excel). Please ensure that all data files are uploaded as 'Supporting Information' and are invariably referred to (in the manuscript, figure legends, and the Description field when uploading your files) using the following format verbatim: S1 Data, S2 Data, etc. Multiple panels of a single or even several figures can be included as multiple sheets in one excel file that is saved using exactly the following convention: S1_Data.xlsx (using an underscore).

b. Deposition in a publicly available repository. Please also provide the accession code or a reviewer link so that we may view your data before publication. 

5) CODE: Per journal policy, if any code was generated to support the conclusions of your manuscript, we would require that you make it available without restrictions upon publication. Please ensure that any code is sufficiently well documented and reusable, and that your Data Statement in the Editorial Manager submission system accurately describes where your code can be found.

REVIEWS:

Reviewer #1: In this study, the authors derive a causal order of events to explain an organ reprogramming paradigm, in which early exposure of fly embryos to ether results in homeotic transformations of halteres to wings later in development. For this, they profile mRNAs, active (trx-catalyzed H3K4me3) and inactive (E(z)-catalyzed H3K27me3) epigenetic histone modifications, and the proteome at various development time points, complementing these genomic analyses with genetic experiments. They provide evidence to support the following conclusions: 

1. Ether appears to induce proteotoxic stress notably of Hsp90 chaperone targets, and somewhat decreases Trx protein levels (Fig. 2). 

2. Ether-induced haltere-to-wing transformation is aggravated by reducing Trx and/or Hsp90 chaperone function (Fig. 3). 

3. Ether exposure leading to partial haltere-to-wing transformations reduces H3K4me3 genome-wide (possibly through Trx protein reduction), though genes related to wing development retain relatively high H3K4me3 levels (Fig. 1). 

4. Genes in embryos that maintained high H3K4me3 levels after early exposure to ether, that are wing and Ubx-trx target genes, become upregulated in larval haltere discs (Figs. 4-5).

The generated data seems of high quality. Significant mechanistic insight was obtained to advance the understanding of how an organ's identity can be transformed by environmental disruption of epigenetic memory. I list minor comments below to further improve the manuscript for publication.

1. RNA-seq performed 1 hour after ether exposure of early embryos did not reveal a strong transcriptional response. This was interpreted to mean that H3K4me3 predisposes genes for a transcriptional response later in development. However, the authors should consider the likely possibility that 1 hour is insufficiently long to see a shift in steady-state mRNA levels (measuring changes in nascent transcription may have revealed a stronger transcriptional response).

2. I believe that the order in which the figures and authors' conclusions are presented could be improved for greater clarity. See the overview in which I propose a possibly more intuitive flow of arguments. 

3. Are Trx protein levels reduced upon genetic or chemical perturbation of Hsp90? Showing this could strengthen the proposed relationship of Trx as a substrate of Hsp90.

4. Section 1 title: "Ether suppresses H3K4me3 trimethylation" should be "H3K4 trimethylation".

5. Fig. 1A: please indicate the time point for the proteomics experiment. I believe this information was also missing from the Methods.

6. Fig. 1: please show a couple of screenshots of K4me3 profiles at example loci, to complement the global analyses presented in this figure.

7. Fig. 1J: please clarify whether the plotted data corresponds to the ether-treated or control conditions.

8. Fig. 2A: please specify in the figure legend or main text how Trx protein levels were quantified.

9. Fig. 2F: please swap the order of the microscopy images, placing the Ctrl on top.

10. Fig. 2: please clarify what happens to protein levels of Ubx, in addition to quantifying Trx protein levels (in Fig. 2A). This remains an important point to clarify - is upregulation of wing genes in halteres of animals exposed to ether early in development caused by decreased Ubx protein levels or to an inability of Ubx to repress its target genes?

11. The meaning of "reaction" was unclear in the sentence: "it suggests that the increase in bithorax reaction in embryos with reduced Hsp90...".

12. Fig. 3D: from the x axis label it is unclear that the last column corresponds to double heterozygotes.

13. There is an incomplete sentence in the first paragraph of the section "Ether exposure up-regulates wing-related Ubx-trx targets in the haltere".

14. "Upregulation of wing genes in the haltere correlates with H3K4me3 levels at the time of exposure": "shortly after exposure" would be more accurate than "at the time of exposure" because it is assessed after exposure, not before. "At the time of exposure" is also repeated in the Discussion.

15. There are several wrong references to figure panels, as in the couple of examples listed below:

a. Fig. 2 legend: references to figure panels are mixed up

b. The reference to Fig. 6A in the Supplementary discussion is wrong.

16. Please specify how many biological replicates were generated for all genome- or proteome-wide datasets.

Reviewer #2: The authors revisit the observation by Waddington in 1956 that Drosophila embryos exposed to ether vapor phenocopy bithorax mutations, with haltere-to-wing transformations. The authors make an impressive connection after they notice that ether exposure compromises the integrity of the eggshell. From there, they suggest that ether leads to global protein misfolding, titrating the chaperone Hsp90 away from misfolded Trithorax. Decreased functional Trithorax reduces H3K4me3 at gene targets, especially those that are also Ubx targets and wing genes. This pathway leads to misexpression of wing genes in the haltere and the resulting phenotypic transformation. 

The authors present compelling data in support of their model. Importantly, their model also accounts for observations about embryonic lethality related to the timing of ether exposure and other bithorax phenocopy observations, including both Hsp90 and trx loss-of-function mutations. 

Overall, this is an impressive and novel story that involves many well-known players of broad interest, such as Hsp90, trx, and H3K4me3. This work points out that there are still many mysteries to be solved, even in the well-studied Drosophila wing system.

We have only minor suggestions to the text and figures to improve the readability and clarity of the manuscript. These are not essential, but are recommended.

The Introduction is extremely short, especially for a study that is revisiting historical observations. Introduction of the historical mystery (i.e. the "Waddington Revisited" portion of the manuscript title), is only briefly mentioned in the first paragraph of the Introduction. The first several paragraphs of the Discussion would be more helpful in the Introduction, as well as more attention paid to the original Waddington discovery and any subsequent follow up (or lack thereof). 

In the first section of the Results, the authors jump from phenotypic observations to ChIP-seq for H3K4me3 and H3K27me3, without explaining why they chose these marks. It is apparent later, with the introduction of trx, but not at this point in the Results section. This logic leap was very distracting when reading the manuscript. Can the authors fill in the logic here?

The narrative jumps around, forming the model in steps that are not linear. From protein misfolding to epigenetic misregulation, then back to protein misfolding and Hsp90. It's not clear in this narrative (until the model) that Hsp90 is operating upstream of trx. The authors can make this clearer in the narrative by building the model sequentially with the figures and introducing the histone modifications later.

Several comments on figures that are all minor:

Fig 1: 

B- The halter is difficult to see. We suggest a blow-up image of this region.

I- Figure legend lists red for up-regulated GO terms, rather than yellow.

K- Some problems with the text

L- Define CDF (y axis)

Fig 2: 

A- Relative to what? 

B- Something is wrong with the legend, as B is not accurately represented 

D- Define "MDEG*ML/MG"

G- Keep nomenclature consistent and replace hsp83 with Hsp90 

Statistics required?

H- In legend "Same as (F)" should read "same as (G)"

Fig 3: 

A-The haltere is too small to see.

A-D are all similar data. Why are B and C prese

---

## [Editor Report · Decision Letter 2]

16 Apr 2024

Dear Orli,

Thank you for the submission of your revised Research Article "Organ transformation by environmental disruption of protein integrity and epigenetic memory in Drosophila" for publication in PLOS Biology. Your revised manuscript has now been assessed by the PLOS Biology editorial staff, and by the Academic Editor, Maria Cristina Gambetta. I am please do say that we are fully satisfied by the changes made in response to the previous reviews and that we can, in principle, accept your manuscript for publication, provided you address any remaining formatting and reporting issues. These will be detailed in an email you should receive within 2-3 business days from our colleagues in the journal operations team; no action is required from you until then. Please note that we will not be able to formally accept your manuscript and schedule it for publication until you have completed any requested changes.

**IMPORTANT: We do have a few last editorial comments and requests, detailed here, which will need to be addressed alongside any formatting requests to come: 

1) Thank you for providing updated supplemental figures containing the uncropped western blots and gels, and the underlying data related to supplemental figures 4-8. As discussed over email, I have updated your submission to include the files that you sent me. Please do look through your submission and at these updated files to make sure everything looks good to you. 

2) Please add a sentence to the figure legends accompanying supplemental figures 4-8, to reference the relevant underlying data files. 

3) Please also update your data availability statement, in our online system, to include a reference to the supplemental data files. For example, you could add the sentence "All other relevant data are included in the manuscript and its supplementary files.

4) We like the new title that you have provided, but would like to suggest a tweak to make it more active. If you agree, we suggest you reorder the title to read: 

"Environmental disruption of protein integrity and epigenetic memory causes organ transformation in Drosophila"

We will ultimately defer to you to make this change, or not. 

PRESS

Sincerely, 

Luke

Lucas Smith, Ph.D.

Senior Editor

PLOS Biology

lsmith@plos.org